# Prevalence of Poor Diet Quality and Associated Factors Among Older Adults from the Bagé Cohort Study of Ageing, Brazil (SIGa-Bagé)

**DOI:** 10.3390/geriatrics10020044

**Published:** 2025-03-17

**Authors:** Tainã Dutra Valério, Rosália Garcia Neves, Elaine Thumé, Karla Pereira Machado, Elaine Tomasi

**Affiliations:** 1Postgraduate Program in Epidemiology, Federal University of Pelotas, Pelotas 96020-220, Brazil; tomasiet@gmail.com; 2Rio Grande do Sul State Department of Health, Porto Alegre 90119-900, Brazil; rosaliagarcianeves@gmail.com; 3Postgraduate Program in Nursing, Federal University of Pelotas, Pelotas 96020-220, Brazil; elaine.thume@ufpel.edu.br; 4Postgraduate Program in Nutrition, Federal University of Pelotas, Pelotas 96020-220, Brazil; karlamachadok@gmail.com

**Keywords:** diet quality, geriatric nutrition, elderly health, population studies, depression

## Abstract

(1) Background: The accelerated aging of the population raises concerns about the diet of older adults due to its relationship with health and quality of life. This study aimed to investigate the prevalence of poor diet quality and its association with sociodemographic factors and health status among older adults residing in the city of Bagé, located in southern Brazil; (2) Methods: A cross-sectional analysis was conducted using data from the 2016/2017 follow-up of the Bagé Aging Cohort Study (SIGa-Bagé). Diet quality was assessed using the Elderly Diet Quality Index. Descriptive analysis and Poisson regression with robust variance adjustment, based on hierarchical levels, were used to calculate crude and adjusted prevalence ratios with their respective 95% confidence intervals; (3) Results: The sample included 728 older adults (65.7% female; mean age: 77.2 years). Poor diet quality was observed in 41.5% of participants. After adjustment, male sex, black or brown skin color, absence of multimorbidity, and presence of depressive symptoms were significantly associated with poor diet quality; (4) Conclusions: The findings highlight the most vulnerable groups and the need for investments in strategies to promote mental health and healthy eating habits among the older adults, particularly among men and racial minority groups.

## 1. Introduction

Since the 1960s, the number and proportion of older adults have been rapidly increasing worldwide. This phenomenon was initially observed only in high-income countries but has also been accelerating in low- and middle-income countries [1]. Much of this trend is attributed to increased life expectancy [2,3], scientific advancements, improvements in quality of life, and declining fertility rates [2,4,5,6]. It is estimated that between 2020 and 2050, the global older adult population (aged 65 and over) will rise from 727 million (9.6%) to 1.5 billion (16.0%) [6].

Data from Brazil’s most recent national census, conducted in 2022, show that the country reached 22.2 million older adults, representing 10.9% of the population—a 57.4% increase over the last decade compared to 14.1 million (7.4%) in 2010. The southern region of Brazil, where the city of Bagé (RS) is located, has the second-highest proportion of older adults (12.1%), trailing only the southeast (12.2%) [7]. The number of older adults in Brazil is projected to reach 50.9 million (28.5%) by 2050 [8].

The aging population raises concerns about diseases that primarily affect this group, including non-communicable chronic diseases (NCDs) [9,10] such as diabetes, hypertension, Alzheimer’s, dementia, and sarcopenia. The cumulative effects of these conditions, in both the short and long term, are associated with the development of functional disabilities [11,12]. Millions of families face the challenge of caring for older adults with disabilities, leading to high costs for long-term care [11,13], which presents a significant challenge for healthcare systems [14]. In this context, it is crucial to understand the habits of this population, such as diet quality, since nutrition is one of the most important modifiable risk factors affecting health. Adequate nutrition is a key determinant of healthy aging, capable of extending longevity and improving the quality of life for older adults [15].

In older adults, diet quality is influenced by sociodemographic factors and age-related physical changes that affect the digestive and sensory systems [4,16,17], including reduced hunger sensations, which lead to decreased diet quality [15,18,19,20,21]. Studies have demonstrated deficiencies in protein, vitamin, and mineral intake among older adults [19,22]. Such poor-quality diets can lead to a decline in nutritional and functional status by negatively impacting muscle mass, as well as metabolic, endocrine, immune, and cognitive functions, posing a significant health risk [16,18].

This study aimed to investigate the prevalence of poor diet quality and its association with sociodemographic factors and health status among older adults residing in the city of Bagé, located in southern Brazil.

## 2. Materials and Methods

This is a population-based cross-sectional study using data from the follow-up of the Bagé Aging Cohort Study (SIGa-Bagé) [23]. The cohort has a fixed population, and at its inception in 2008, it consisted of a representative sample of individuals aged 60 years or older, of both sexes, non-institutionalized, living in private households, and residing within the coverage area of primary healthcare services in the urban area of Bagé, Rio Grande do Sul. In 2008, at baseline, 1593 older adults were interviewed. During the 2016/2017 follow-up, excluding deaths, losses, and refusals, 735 older adults were re-interviewed, all of whom were 68 years or older at the time. Of these, 728 participants responded to questions about dietary intake, making them eligible for inclusion in the present study.

Poor diet quality (the outcome) was assessed using the Elderly Diet Quality Index (EDQI) [24] based on the frequency of food consumption. For this purpose, a shortened version of a Food Frequency Questionnaire (FFQ) was applied, where older adults reported how many days in the past week they had consumed the following foods or food groups: rice with beans or rice with lentils; whole foods (wholegrain bread, wholegrain cookies, wholegrain rice or oats); vegetables and greens; fruits; red meat, chicken, fish or eggs; milk, yogurt, or cheese; fried foods; candies, sodas and boxed or packaged juices; sausages and hams, pickles (gherkins), and canned foods (sardines or canned fruit and vegetables); frozen foods (lasagna, pizza, hamburgers, and nuggets); and snacks (from food trucks or fast-food outlets) [24].

Each healthy food was scored as follows: did not consume = 0; consumed on 1 to 3 days = 1; consumed on 4 to 6 days = 2; consumed on every day of the week = 3. Each unhealthy food was scored inversely: did not consume = 3; consumed on 1 to 3 days = 2; consumed on 4 to 6 days = 1; consumed on every day of the week = 0. The EDQ-I was calculated by summing the scores for each food or food group together, producing a number between 0 and 33, where the higher the score the greater the frequency of consumption of healthy foods. The overall scores were then divided into the following terciles: 1st—poor dietary quality, score between 14 and 24; 2nd—medium dietary quality, score between 25 and 27; and 3rd—high dietary quality, score between 28 and 33 [24].

The independent variables were: sex (male; female), age (68–79 years; 80 years or older), skin color (white; black; brown), marital status (with a partner/married; without a partner/single), living arrangement (no; yes), education level (no formal education; up to 8 years; 9 years or more), per capita income tertile (≤BRL 602.00; BRL 606.67–BRL 1000.00; ≥BRL 1010.00), use of dental prostheses (no; yes), problems or difficulties chewing or swallowing food (no; yes), need for help with eating (no; yes), need for help with preparing meals (no; yes), body mass index (BMI) (underweight; normal weight; overweight) [25] and multimorbidity—the latter was defined as the presence of at least five [26] of the following self-reported health problems: high blood pressure; diabetes; heart problems; lung problems or COPD, asthma, bronchitis, emphysema; osteoporosis, arthritis, osteoarthritis, or rheumatism; Parkinson’s disease; renal insufficiency; prostrate diseases (in the case of male respondents); thyroid problems; glaucoma; cataracts; Alzheimer’s; angina; heart attack; stroke; high cholesterol; epilepsy or convulsions; depression; stomach or duodenal ulcers; urine infection, urinary incontinence, constipation, bowel incontinence; deafness; insomnia or difficulty sleeping; fainting; difficulty speaking; and cancer. The association between the EDQ-I and depressive symptoms was also assessed using the shorter version of the Geriatric Depression Scale (GDS) [27]. Older adults who scored less than six were classified as having “absence of depressive symptoms”, while those with a score of six or higher were classified as having “presence of depressive symptoms”.

The follow-up interviews were conducted in person at the older adults’ homes between September 2016 and August 2017. The interviewers, all female and over 18 years of age, were trained to administer the questionnaire and perform the physical assessment of the older adults.

Descriptive statistics, including mean, median, and standard deviation (SD), were used. Poisson regression with robust variance estimates was used to calculate crude and adjusted prevalence ratios and respective 95% confidence intervals (95% CI). Variables with a *p*-value of <0.20 were included in the adjusted analysis to determine the effect of each independent variable on the outcome, controlling for stratum-specific confounding adopting a 5% significance level. The Poisson Regression was chosen over Logistic Regression because it is more suitable for cross-sectional studies with high prevalence outcomes, as logistic regression tends to overestimate the effect size when the prevalence of the outcome is greater than 10% [28]. The frequency of weekly consumption of each food group included in the IQD-I was analyzed (prevalence and 95% CI), according to sex and the presence of depressive symptoms. All statistical analyses were performed using Stata/SE 17.0.

Due to its cross-sectional design, this study has some limitations, including the possibility of reverse causality. However, this was mitigated by the temporality of the variables, as all exposure variables refer to periods preceding the outcome. Additionally, there is a risk of recall bias, which was minimized by the short recall period of the questionnaire.

## 3. Results

The sample consisted of 728 older adults, of whom 65.7% were women. The mean age was 77.2 years (SD = 6.5 years), and 69.2% were aged between 68 and 79 years. Most of the individuals were White (82.6%), without a partner/single (57.3%), lived with others (75.6%), and had up to eight years of schooling (60.6%). Approximately one-third of the older adults reported a per capita income of up to BRL 602.00, and the mean income was BRL 1292.97 (SD = BRL 1616.56). The majority used dental prostheses (75.3%), had no difficulty chewing or swallowing food (89.2%), ate without assistance (95.2%), and did not require help preparing meals (81.1%). The presence of five or more morbidities was identified in 45.7% of the population, 13.5% presented depressive symptoms, and 47.5% were overweight (Table 1).

Mean and median EDQ-I scores were 25.2 (SD = 4.0) and 26, respectively, and 41.5% of the sample (95% CI: 37.9–45.1) had poor diet quality (Table 1).

In the crude analysis, the variables sex, age, skin color, per capita income, use of dental prostheses, multimorbidity, and presence of depressive symptoms were significantly associated with the outcome. Poor diet quality was 35% higher among men compared to women (PR = 1.35; 95% CI: 1.14–1.60) and 25% higher in individuals aged 68–79 years compared to those aged 80 years or older (PR = 1.25; 95% CI: 1.02–1.54). Regarding skin color, the prevalence of poor diet quality was 41% (PR = 1.41; 95% CI: 1.12–1.78) and 51% (PR = 1.51; 95% CI: 1.17–1.96) higher among black and brown individuals, respectively, compared to White participants (Table 2).

Older adults with lower per capita income had a 13% higher probability of poor diet quality compared to those with higher income (PR = 1.13; 95% CI: 0.93–1.39). Not using dental prostheses increased the probability of poor diet quality by 27% compared to users (PR = 1.27; 95% CI: 1.06–1.52). Poor diet quality was 20% higher among those without multimorbidity (PR = 1.20; 95% CI: 1.00–1.45) and 34% higher among those with depressive symptoms (PR = 1.34; 95% CI: 1.08–1.67) (Table 2).

In the adjusted analysis, only sex, skin color, multimorbidity, and depressive symptoms remained significantly associated with poor diet quality. Poor diet quality was 30% higher among men (PR = 1.30; 95% CI: 1.08–1.58). Among black and brown participants, the prevalence of poor diet quality was 33% (PR = 1.33; 95% CI: 1.03–1.71) and 44% (PR = 1.44; 95% CI: 1.08–1.94) higher, respectively, compared to White participants. Poor diet quality was 24% higher among those without multimorbidity (PR = 1.24; 95% CI: 1.02–1.50) and 48% higher among those with depressive symptoms (PR = 1.48; 95% CI: 1.17–1.86) (Table 2).

When evaluating the weekly frequency of consumption of each food group included in the EDQ-I, by sex, it was observed that men had a lower intake of whole grains, vegetables, fruits, and dairy products, and a higher consumption of fried foods, preserves and canned foods, frozen foods, and snacks (Table 3).

When evaluating the weekly consumption of foods individually, older adults with depressive symptoms showed a lower intake of healthy foods (rice with beans or rice with lentils; vegetables; fruits; meat, chicken, fish, or eggs; and milk, yogurt, or cheese) and higher consumption of unhealthy foods (fried foods; Candies, sodas and boxed or packaged juices); and frozen foods (lasagna, pizza, hamburgers, and nuggets) (Table 4).

## 4. Discussion

The overall prevalence of poor diet quality in our sample was 41,5%, with rates reported in the literature varying between 14.6% [29] and 44.6% [30]. This variation may be explained by differences in the sociodemographic profile of study samples, access to food, cultural aspects across different countries, and lack of standardized data collection instruments [31]. Despite this variation, there is a consensus among researchers that an unhealthy or poor-quality diet is characterized by high consumption of ultra-processed foods, which contain high levels of fat and refined sugar, and low intake of fruits, vegetables, and greens [2,15,17,21,24,30,32,33,34]. Other authors have used dietary quality indices to investigate dietary quality [34], an example is the EDQ-I adapted by Gomes et al. [15,24] who obtained similar results to those found in Bagé (mean = 24.2; SD = 3.8 versus mean = 25.3; SD = 3.7).

The use of a cross-sectional research design means that the present study has some limitations, including recall bias. However, the effects of this type of bias were minimized due to the short recall periods for some variables. We opted to use diet quality as the outcome variable and depressive symptoms as one of the exposure variables considering the respective recall periods adopted by the measures used to assess these variables. This study is also subject to reverse causality, which may be present in the bidirectional association between depressive symptoms and diet quality [35], as well as in its relationship with multimorbidity.

The prevalence of depressive symptoms was 14.0%, which is similar to the rate found (15.3%) by Gomes et al. [17], and its association between this factor and diet quality has been demonstrated by other authors [17,21,32,35,36]. High-quality diets are consistently associated with a lower risk of depressive symptoms [37,38,39].

Nutrient deficiencies and even malnutrition are common in older adults [18], with low levels of specific nutrients such as folic acid, vitamin B12 and omega-3 being found in older adults with depression [40]. Similar findings were observed among the older adult population in Bagé, where an evaluation of the weekly frequency of isolated food group consumption included in the EDQ-I showed that older adults with depressive symptoms had a lower intake of healthy foods and a higher consumption of fried and frozen foods.

The potentially bidirectional relationship between depressive symptoms and diet quality has also been investigated, as these variables go hand in hand, with the exposure variable often acting as an outcome variable and vice versa. Jacka et al. [35] suggested that the relationship between both short- and long-term diet and depression is complex, with evidence indicating that poor-quality diet increases the risk of depression and anxiety and that better-quality diets act as a protective factor [36]. At the same time, evidence also suggests that, in the short term, the consumption of sugary foods helps people with mood disorders alleviate distress. However, in a study investigating reverse causality in the relationship between diet quality and depression, testing the hypothesis that prior depression is associated with poorer diet quality at a later time point, the same authors conclude that a history of depression may prompt healthier dietary behaviors in the long term, perhaps as an attempt to improve residual symptoms or prevent future episodes of depression [35].

Unlike the findings of Gomes et al. [41] and Pereira et al. [15], the present study identified an association between poor diet quality and multimorbidity. Similar results were observed by Mayer [42], who, analyzing data from the 2019 National Health Survey (PNS), found a positive association between irregular bean consumption and multimorbidity. Conversely, insufficient consumption of fruits, vegetables, and leafy greens, as well as meal replacement with snacks, appeared to be protective factors against multimorbidity. However, in the present sample, this relationship may have been influenced by reverse causality, as the prevalence of poor diet quality was 24% higher among older adults without multimorbidity (PR = 1.24; 95% CI: 1.02–1.50). A possible explanation for this finding is that individuals with multimorbidity tend to access healthcare services more frequently due to the management of their conditions, thereby being more regularly exposed to nutritional counseling and recommendations for healthy behaviors. Consequently, these individuals may be engaged in a dietary re-education process aimed at managing their health conditions.

Poor diet quality is a modifiable risk factor for several health conditions, such as non-communicable chronic diseases (NCDs), sarcopenia, malnutrition, and obesity, which, in turn, act as risk factors for functional disability [12,43,44,45,46,47,48]. Although the present study did not find an association between poor diet quality and BMI, other authors have demonstrated that healthy dietary patterns are inversely associated with the risk of obesity and overweight [49]. They are also associated with lower cognitive impairment [38,39,50], reduced chances of physical frailty [51,52] and lower mortality rates [53].

A higher prevalence of poor dietary quality in men was also found by Araújo et al. [31] in a review of 18 original national articles published between 2011 and 2020. This may be partially explained by the fact that women have traditionally acted as the main caregivers in the family, meaning they concern themselves more with selecting foods and preparing meals [18,30,31,33,41,54]. On the other hand, men are less likely to seek and utilize health services, where they would be exposed to activities promoting healthy eating habits [24,30].

In our study, we observed that men had a lower intake of whole grains, vegetables, fruits, and dairy products, and a higher consumption of meats, fried foods, canned and preserved foods, frozen meals, and snacks. It is also important to highlight the taboos around food culture in Brazil, where red meat is linked with notions of masculinity, virility, and prosperity, especially in the south of Rio Grande do Sul, and fruits and vegetables are considered more suitable for children, women, and older people and generally viewed as unnecessary items in times of financial hardship [33].

Studies have shown that diet quality is also associated with demographic and socioeconomic characteristics [21,30,31,55], and it can even be influenced by food insecurity, which primarily affects the most vulnerable populations, such as those with lower income, lower education levels, and black and brown individuals [56]. This same racial disparity in the consumption of healthy foods was observed in a study by Figueiredo Neta [57] using data from the National Health Survey 2019. The study found that the main disparity in food consumption was related to fruit intake among the Black population (PR = 0.91; 95% CI: 0.88–0.95), even after adjusting for socioeconomic and demographic variables.

Our findings also showed an association between diet quality and skin color, with the prevalence of poor diet quality being higher in black and brown respondents. This may be explained by social inequalities and lack of access to information and nutrient-rich foods among the former two groups, given that 43.4% of black respondents and 53.8% of brown respondents were in the first income tercile, according to supplementary analysis. Demonstrating the presence of an association between worse social conditions, lifestyle habits, and poor diet quality, reinforcing the hypothesis of social determination and the coexistence of health behaviors [30].

This study adhered to the STROBE guidelines for observational studies, ensuring methodological rigor. Its findings can be generalized to non-institutionalized older adults living in medium-sized municipalities in Brazil. However, it is essential to consider potential regional and socioeconomic variations that may affect the applicability of the results in other contexts.

## 5. Conclusions

The overall prevalence of poor diet quality in our sample was 41%. Prevalence was higher in men, black and brown older adults, those without multimorbidity, and those with depressive symptoms. Our findings reveal the need for greater investment in strategies to promote mental health and healthy eating habits among the older population, especially among men and racial minority groups. Future research should focus on longitudinal studies to investigate the relationship between depression and diet quality.

## Figures and Tables

**Table 1 geriatrics-10-00044-t001:** Description of the sample according to demographic, socioeconomic characteristics, and health status of the global older adults population of Bagé. Bagé, Rio Grande do Sul, 2016/2017 (*n* = 728).

Variables	Total *n* (%)
**Sex**	
Female	478 (65.7)
Male	250 (34.3)
**Age (years)**	
68 a 79	504 (69.2)
80 or more	224 (30.8)
**Skin color ^3^**	
White	598 (82.6)
Black	76 (10.5)
Brown	50 (6.9)
**Marital status ^3^**	
With a partner/married	309 (42.7)
Without a partner/married	415 (57.3)
**Lives alone**	
No	550 (75.6)
Yes	178 (24.4)
**Education level (years completed) ^3^**	
No education	161 (22.3)
Up to 8	438 (60.6)
9 or more	124 (17.1)
**Income tercile (BRL) ^3^**	
1st (≤602.00)	240 (33.2)
2nd (606.67–1000.00)	250 (34.5)
3rd (≥1010.00)	234 (32.3)
**Use of dental prothesis ^3^**	
No	178 (24.7)
Yes	544 (75.3)
**Difficulty chewing or swallowing**	
No	649 (89.2)
Yes	79 (10.8)
**Needs help with eating ^3^**	
No	692 (95.2)
Yes	35 (4.8)
**Needs help with preparing meals ^3^**	
No	588 (81.1)
Yes	137 (18.9)
**Multimorbidity ^1,3^ (≥5 morbidities)**	
No	383(54.3)
Yes	322 (45.7)
**Depressive symptoms ^2,3^ (GDS ≥ 6)**	
No	598 (86.5)
Yes	93 (13.5)
**Body Mass Index ^3^ (Lipschitz)**	
Underweight	109 (16.4)
Normal weight	240 (36.1)
Overweight	316 (47.5)
**Elderly Dietary Quality Index (EDQ-I)**	
Poor (score ≤ 24)	302 (41.5)
Medium (score 25–27)	205 (28.2)
High (score ≥ 28)	221 (30.3)

^1^ Defined as the presence of at least five of the following self-reported morbidities: high blood pressure; diabetes; heart problems; lung problems or DPOC, asthma, bronchitis, emphysema; osteoporosis, arthritis or osteoarthritis, rheumatism; Parkinson’s disease; renal insufficiency; prostrate diseases (in the case of male respondents); thyroid problems; glaucoma; cataracts; Alzheimer’s; angina; heart attack; stroke; high cholesterol; epilepsy or convulsions; depression; stomach or duodenal ulcers; urine infection; urinary incontinence; constipation; bowel incontinence; deafness; insomnia or difficulty sleeping; fainting; difficulty speaking; and cancer. ^2^ DGS Geriatric Depression Scale [27]. ^3^ Missing data Skin color (*n* = 4); Marital status (*n* = 4); Education level (*n* = 5); Use of dental prothesis (*n* = 4); Needs help with eating (*n* = 1); Needs help with preparing meals (*n* = 3); Multimorbidity (*n* = 23); Depressive symptoms (*n*=37); Body Mass Index (*n* = 63).

**Table 2 geriatrics-10-00044-t002:** Prevalence, crude and adjusted analyses between poor diet quality and demographic, socioeconomic, and health characteristics. Bagé, Rio Grande do Sul, 2017 (*n* = 294).

Variables	Poor Diet Quality		
	Prevalence%	Crude PR ^1^ (95% CI)	Adjusted PR ^2^ (95% CI)
**Sex**		*p* < 0.001	*p* = 0.007
Female	37.0	1	1
Male	50.0	1.35 (1.14–1.60)	1.30 (1.08–1.58)
**Age (years)**		*p* = 0.028	*p* = 0.068
68 a 79	44.3	1.25 (1.02–1.54)	1.20 (0.99–1.47)
80 or more	35.3	1	1
**Skin color**		*p* < 0.001	*p* = 0.010
White	38.3	1	1
Black	54.0	1.41 (1.12–1.78)	1.33 (1.03–1.71)
Brown	58.0	1.51 (1.17–1.96)	1.44 (1.08–1.94)
**Marital status**		*p* = 0.589	-
With a partner/married	42.7	1.05 (0.88–1.25)	-
Without a partner/married	40.7	1	-
**Lives alone**		*p* = 0.243	-
No	42.7	1.14 (0.92–1.40)	-
Yes	37.6	1	-
**Education level (years completed)**		*p* = 0.179	*p* = 0.405
No education	35.4	1	1
Up to 8	43.8	1.24 (0.98–1.56)	0.82 (0.65–1.03)
9 or more	39.5	1.12 (0.83–1.51)	0.90 (0.69–1.16)
**Income tercile (BRL)**		*p* = 0.023	*p* = 0.081
1st (≤602.00)	47.5	1.13 (0.93–1.39)	1.09 (0.88–1.37)
2nd (606.67–1000.00)	35.2	0.84 (0.67–1.05)	0.85 (0.67–1.08)
3rd (≥1010.00)	41.9	1	1
**Use of dental prosthesis**		*p* = 0.011	*p* = 0.135
No	49.4	1.27 (1.06–1.52)	1.17 (0.95–1.43)
Yes	39.0	1	1
**Difficulty chewing or swallowing**		*p* = 0.287	-
No	40.8	1	-
Yes	46.8	1.15 (0.89–1.48)	-
**Needs help with eating**		*p* = 0.865	-
No	41.5	1.04 (0.68–1.57)	-
Yes	40.0	1	-
**Needs help with preparing meals**		*p* = 0.747	-
No	41.7	1.04 (0.83–1.30)	-
Yes	40.2	1	-
**Multimorbidity ^3^ (≥5 morbidities)**		*p* = 0.046	*p* = 0.030
No	44.1	1.20 (1.00–1.45)	1.24 (1.02–1.50)
Yes	36.7	1	1
**Body Mass Index (Lipschitz)**		*P* = 0.213	-
Underweight	43.1	0.98 (0.76–1.26)	-
Normal weight	36.7	1	-
Overweight	44.0	0.83 (0.68–1.03)	-
**Depressive symptoms (GDS ≥ 6)**		*p* = 0.008	*p* <0.001 ^4^
No	39.3	1	1
Yes	52.7	1.34 (1.08–1.67)	1.48 (1.17–1.86)

^1^ Crude prevalence ratio. ^2^ Adjusted at each level of the three-strata hierarchical model: 1st sex, age and skin color; 2nd per capita income; 3rd use of dental prostheses, multimorbidity, body mass index, and depressive symptoms. Only variables with *p*-value < 0.20 were included in each stratum. ^3^ Defined as the presence of at least five of the following self-reported morbidities: high blood pressure; diabetes; heart problems; lung problems or DPOC, asthma, bronchitis, emphysema; osteoporosis, arthritis or osteoarthritis, rheumatism; Parkinson’s disease; renal insufficiency; prostrate diseases (in the case of male respondents); thyroid problems; glaucoma; cataracts; Alzheimer’s; angina; heart attack; stroke; high cholesterol; epilepsy or convulsions; depression; stomach or duodenal ulcers; urine infection; urinary incontinence; constipation; bowel incontinence; deafness; insomnia or difficulty sleeping; fainting; difficulty speaking; and cancer. ^4^ Adjusted for sex, skin color, per capita income, use of dental prostheses, and multimorbidity.

**Table 3 geriatrics-10-00044-t003:** Weekly frequency of isolated consumption of each food group included in the EDQ-I, by sex. Bagé, RS, 2016/2017 (*n* = 624).

Food	Female (*n* = 408) Prevalence (95% CI)				Male (*n* = 216) Prevalence (95% CI)			
	Did Not Consume	1–3 Days	4–6 Days	Every Day of the Week	Did Not Consume	1–3 Days	4–6 Days	Every Day of the Week
**Healthy food**								
Rice with beans or rice with lentils	5.3 (3.6–7.7)	14.3 (11.5–17.8)	13.5 (10.7–16.9)	66.9 (62.5–71.0)	2.4 (1.1–5.3)	6.8 (4.3–10.7)	19.3 (14.8–24.7)	71.5 (65.5–76.8)
Whole foods (wholegrain bread, wholegrain cookies, wholegrain rice or oats)	52.6 (48.1–57.1)	13.3 (10.5–16.7)	10.6 (8.1–13.7)	23.5 (19.9–27.5)	65.3 (59.2–71.0)	15.3 (11.3–20.4)	2.8 (1.3–5.8)	16.5 (12.4–21.7)
Vegetables and greens	7.9 (5.8–10.8)	21.6 (18.1–25.5)	20.1 (16.8–24.0)	50.3 (45.8–54.8)	9.2 (6.2–13.5)	24.1 (19.2–29.8)	23.7 (18.8–29.4)	43.0 (36.9–49.2)
Fruits	6.1 (4.2–8.6)	15.2 (12.3–18.8)	17.3 (14.2–21.0)	61.4 (56.9–65.6)	13.3 (9.6–18.1)	16.5 (12.3–21.6)	18.9 (14.5–24.2)	51.4 (45.2–57.6)
Red meat, chicken, fish or eggs	2.1 (1.1–3.9)	9.9 (7.5–13.0)	16.5 (13.4–20.1)	71.5 (67.2–75.4)	0.8 (0.2–3.2)	7.7 (4.9–11.7)	16.1 (12.0–21.3)	75.4 (69.6–80.4)
Milk, yogurt, or cheese	10.3 (7.8–13.3)	15.7 (12.7–19.2)	11.7 (9.1–14.9)	62.3 (57.9–66.6)	16.8 (12.6–22.0)	16.0 (11.9–21.1)	12.0 8.5–16.7)	55.2 (49.0–61.3)
**Unhealthy food**								
Fried foods	64.9 (60.5–69.1)	30.4 (26.5–34.8)	2.7 (1.6–4.7)	1.9 (1.0–3.6)	52.4 (46.2–58.6)	40.3 (34.4–46.6)	2.8 (1.3–5.8)	4.4 (2.5–5.8)
Candies, sodas and boxed or packaged juices	34.7 (30.1–39.1)	31.7 (27.7–36.1)	10.4 (7.9–13.5)	23.3 (19.7–27.3)	30.6 (25.2–36.7)	35.5 (29.7–41.7)	13.3 (9.6–18.2)	20.6 (16.0–26.1)
Sausages and hams, pickles (gherkins), and canned foods (sardines or canned fruit and vegetables)	76.6 (72.5–80.2)	18.6 (15.3–22.3)	3.8 (2.4–6.0)	1.1 (0.4–2.5)	67.3 (61.2–72.9)	23.4 (18.5–29.1)	6.9 (4.3–10.8)	2.4 (1.1–5.3)
Frozen foods (lasagna, pizza, hamburgers, and nuggets)	90.1 (87.1–92.5)	9.3 (7.0–12.2)	0.6 (0.2–1.9)	-	88.3 (83.7–91.8)	11.3 (7.9–15.9)	0.4 (0.1–2.8)	-
Snacks (from food trucks or fast-food outlets)	94.7 (92.3–96.4)	5.2 (3.6–7.7)	-	-	93.1 (89.2–95.7)	6.9 (4.3–10.8)	-	-

**Table 4 geriatrics-10-00044-t004:** Weekly frequency of isolated consumption of each food group included in the EDQ-I, according to the presence of depressive symptoms. Bagé, RS, 2016/2017 (*n* = 624).

Food	Absence of Depressive Symptoms (*n* = 536) Prevalence (95% CI)				Presence of Depressive Symptoms (*n* = 88) Prevalence (95% CI)			
	Did Not Consume	1–3 Days	4–6 Days	Every Day of the Week	Did Not Consume	1–3 Days	4–6 Days	Every Day of the Week
**Healthy food**								
Rice with beans or rice with lentils	3.8 (2.6–5.7)	11.4 (9.1–14.2)	16.2 (13.5–19.4)	68.6 (64.7–72.2)	5.4 (2.2–12.4)	14.0 (8.2–22.7)	12.9 (7.4–21.5)	67.7 (57.5–76.5)
Whole foods (wholegrain bread, wholegrain cookies, wholegrain rice or oats)	55.4 (51.3–59.3)	14.2 (11.6–17.3)	8.7 (6.7–11.2)	22.7 (18.6–25.2)	64.1 (53.7–73.4)	16.3 (10.0–25.4)	4.3 (1.6–11.1)	15.2 (9.2–24.2)
Vegetables and greens	7.4 (5.5–9.8)	22.6 (19.4–26.2)	21.3 (18.2–24.7)	48.7 (44.7–52.8)	14.1 (8.3–23.0)	26.1 (18.0–36.1)	20.7 (13.5–30.3)	39.1 (29.6–49.6)
Fruits	7.7 (5.8–10.1)	15.2 (12.6–18.3)	18.6 (15.6–21.9)	58.5 (54.5–62.4)	11.8 (6.6–20.2)	18.3 (11.6–27.6)	16.1 (9.9–25.2)	53–8 (43.5–63.7)
Red meat, chicken, fish or eggs	1.2 (0.6–2.4)	8.9 (6.8–11.4)	15.6 (12.9–18.7)	74.4 (70.8–77.8)	3.2 (1.0–9.7)	9.7 (5.1–17.7)	21.5 (14.2–31.1)	65.6 (55.3–74.6)
Milk, yogurt, or cheese	12.4 (10.0–15.3)	16.1 (13.3–19.2)	11.5 (9.2–14.4)	60.0 (56.0–63.9)	14.0 (8.2–22.7)	19.4 (12.5–28.8)	9.7 (5.1–17.7)	57.0 (46.6–66.8)
**Unhealthy food**								
Fried foods	60.7 (56.7–64.6)	33.9 (30.2–37.8)	2.7 (1.6–4.3)	2.7 (1.6–4.3)	59.1 (48.8–68.8)	33.3 (24.4–43.6)	4.3 (1.6–11.0)	3.2 (1.0–9.7)
Candies, sodas and boxed or packaged juices	32.7 (29.0–36.5)	33.7 (30.0–37.6)	11.7 (9.4–14.6)	21.9 (18.8–25.4)	28.0 (19.7–38.0)	31.2 (22.5–41.4)	11.8 (6.6–20.2)	29.0 (20.6–39.2)
Sausages and hams, pickles (gherkins), and canned foods (sardines or canned fruit and vegetables)	72.1 (68.3–75.5)	21.2 (18.1–24.7)	5.4 (3.8–7.5)	1.3 (0.6–2.7)	78.5 (68.9–85.8)	16.1 (9.9–25.2)	3.2 (1.0–9.7)	2.2 (0.5–8.3)
Frozen foods (lasagna, pizza, hamburgers, and nuggets)	89.5 (86.7–91.7)	9.9 (7.7–12.5)	0.7 (0.3–1.8)	-	89.2 (81.0–94.2)	10.8 (5.8–19.0)	-	-
Snacks (from food trucks or fast-food outlets)	93.6 (91.4–95.3)	6.4 (4.7–8.6)	-	-	96.8 (90.3–99.0)	3.2 (1.0–9.7)	-	-

## Data Availability

The raw data supporting the conclusions of this article will be made available by the authors upon request.

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
