# Peer review of "Prevalence of Poor Diet Quality and Associated Factors Among Older Adults from the Bagé Cohort Study of Ageing, Brazil (SIGa-Bagé)"

_geriatrics, 2025, doi:10.3390/geriatrics10020044_

Round 1

Reviewer 1 Report (Previous Reviewer 1)

Comments and Suggestions for Authors I am happy with the changes.

Author Response

I would like to express my sincere gratitude for the time and effort you dedicated to reviewing my article.

Reviewer 2 Report (New Reviewer)

Comments and Suggestions for Authors

Summary

The manuscript examines the prevalence of poor diet quality and its associated factors among older adults in Brazil. Using data from the SIGa-Bagé cohort, it identifies significant associations between poor diet quality and variables such as sex, race, multimorbidity, and depressive symptoms. The study employs Poisson regression analysis to estimate prevalence ratios.

    The authors must explicitly state their adherence to the STROBE guidelines for observational studies and ensure that all relevant checklist items are addressed in the manuscript​.

    The manuscript states that a short version of an FFQ was used to assess diet quality​. However, there is no information about whether this FFQ has been previously validated or used in scientific literature. The authors must clarify its validation status to ensure that the dietary assessment is reliable.

    Reference 24, which is cited for the Elderly Diet Quality Index (EDQI), is inaccessible and not in English​. Since the EDQI is central to the study, the authors must provide a more detailed explanation of its validation and measurement properties to justify its use.

    The discussion lacks depth regarding the reasons behind the associations observed. For example, why is the absence of multimorbidity linked to poor diet quality? Why do Black or mixed-race individuals and those with depressive symptoms exhibit worse diet quality? Without theoretical or empirical explanations, the discussion remains superficial and does not contribute significantly to understanding these relationships​.

    The use of Poisson regression with robust variance needs justification. While it is sometimes used for prevalence ratio estimation in cross-sectional studies, logistic regression or other multivariate methods are more commonly applied. The authors should explain why they chose Poisson regression over logistic regression​.

    The manuscript uses the term "elderly," which is often considered outdated or ageist. The authors should replace it with "older adults" or "older individuals" to align with current best practices in gerontology and public health research​.

    The manuscript does not clearly explain the relevance of Tables 3 and 4 in addressing the study's objectives. If these tables do not provide essential insights, the authors should justify their inclusion or consider removing them to improve clarity and conciseness​.

Comments on the Quality of English Language

The text must be reviewed by a native Enghish speaker

Round 2

Reviewer 2 Report (New Reviewer)

Comments and Suggestions for Authors

OK

This manuscript is a resubmission of an earlier submission. The following is a list of the peer review reports and author responses from that submission.

Round 1

Reviewer 1 Report

Comments and Suggestions for Authors

This is a very good paper and certainly reflects the issues regarding food consumption and a dearth of nutrients which affects the ageing population as it grows worldwide. 

The protein, vitamin and mineral deficiency is of paramount importance worldwide. Data was used from the Bage study, as a follow up. Interviews conducted in the persons home. Perhaps a suggestion might be to link the specific dearth in certain foods to depression would add weight to the paper. (Line 163). Evidence should also be drawn into the conclusion.

Reviewer 2 Report

Comments and Suggestions for Authors

I have included some memos within the pdf file. 

With the intention of this study, I expected much more in-depth analyses. 

Only two tables seem not enough. 

However, apart from that, the authors gave minimal discussion regarding the results. For example, it seems that the younger elderly were at greater risk for poor dietary quality, and that was not even mentioned. Living with a partner also posed a higher risk for poor diet quality. The authors did not give much thought to that. 

Reviewer 3 Report

Comments and Suggestions for Authors

The manuscript investigates the prevalence of poor diet quality and its associations with sociodemographic and health factors among older adults in Bagé, Brazil. The study is significant given the aging population and the importance of diet in maintaining health and quality of life in older adults. The methodology is sound, the data analysis is appropriate, and the findings are relevant. However, several areas require improvement and additional context to strengthen the manuscript.

Major points for revision

Introduction

Enhance the introduction by providing a more detailed background on the aging population in Brazil and globally. Include statistics on the prevalence of chronic diseases related to diet quality in older adults.

Methods

Clarify the selection criteria for the sample and any potential biases introduced by the inclusion/exclusion criteria.

Results

Provide more detailed subgroup analyses to explore if and how different factors (e.g., income, education) interact to affect diet quality.

Discussion

Deepen the discussion of potential interventions to improve diet quality among the identified vulnerable groups. Include a comparison with other similar studies worldwide to highlight the uniqueness or commonality of your findings. More specifically:

1.  The manuscript should address the possible implications of poor dietary quality and unhealthy dietary patterns, including the higher risk of obesity and metabolic syndrome (https://pubmed.ncbi.nlm.nih.gov/38068751/), cognitive decline (https://pubmed.ncbi.nlm.nih.gov/38158186/) and decreased physical performance and sarcopenia. These conditions are prevalent in older adults and have significant health impacts. Discussing these implications would provide a more comprehensive understanding of the consequences of poor dietary quality.

2.   The manuscript mentions the association between diet quality and various factors such as sex, skin color, and depressive symptoms. However, it would benefit from a discussion on the evidence of clusterization of poor dietary quality with other health and sociodemographic factors. For instance, socioeconomic status, access to healthcare, and educational background often cluster with dietary habits and can contribute to overall health disparities.

3. Expand the discussion on the factors contributing to worse overall health outcomes among those with poor diet quality. This includes not only depressive symptoms but also physical health conditions, social isolation, and functional limitations.

4.  Recommend possible healthy dietary models for older adults that may be sustainable both economically and environmentally.

Conclusion 

The conclusion should emphasize the need for policy and community interventions targeting the identified vulnerable groups. Highlight the importance of mental health services in conjunction with dietary interventions.

Minor Points for Revision

1. Ensure the manuscript is free from grammatical errors and awkward phrasing.

2. The section on ethical considerations is sufficient, but it would benefit from a brief mention of how informed consent was obtained, particularly from vulnerable populations.

The manuscript offers valuable insights into the diet quality of older adults in Bagé, Brazil. With the suggested revisions, it can significantly contribute to the literature on geriatric nutrition and public health interventions. I recommend a major revision to address the highlighted points and improve the overall quality and impact of the study.

Comments on the Quality of English Language

Ensure the manuscript is free from grammatical errors and awkward phrasing.